# The Effect of Humic Mineral Substances from Oxyhumolite on the Coagulation Properties and Mineral Content of the Milk of Holstein-Friesian Cows

**DOI:** 10.3390/ani11071970

**Published:** 2021-06-30

**Authors:** Anna Teter, Monika Kędzierska-Matysek, Joanna Barłowska, Jolanta Król, Aneta Brodziak, Mariusz Florek

**Affiliations:** Institute of Quality Assessment and Processing of Animal Products, Faculty of Animal Sciences and Bioeconomy, University of Life Sciences in Lublin, Akademicka 13, 20-950 Lublin, Poland; monika.matysek@up.lublin.pl (M.K.-M.); joanna.barlowska@up.lublin.pl (J.B.); jolanta.krol@up.lublin.pl (J.K.); aneta.brodziak@up.lublin.pl (A.B.); mariusz.florek@up.lublin.pl (M.F.)

**Keywords:** humic substances, milk yield, milk clotting time, rennet curd, texture, curd firmness

## Abstract

**Simple Summary:**

Global commodity milk production relies mainly on Holstein-Friesian cows. Intensive selection has made this breed highly productive, but deterioration of health, reduced longevity, and problems with milk quality have been observed. Much of the milk produced is used for cheese production, where its coagulation capacity is crucial. Dietary modifications are used to improve the health and milk quality of cows. Recent years have seen growing interest in humic substances in animal diets. Humic substances are natural organic substances formed in soil during humification of dead organic matter. Their main components are humic acids, fulvic acids and humins. Moreover, humic substances are a rich source of easily absorbed minerals. They are considered natural and safe feed additives with a number of positive effects, including improve of animal welfare and the quality of animal products. The literature lacks information on the results of the use of humic mineral substances in the diet of cows on the suitability of milk for cheese production. Therefore, the aim of the study was to assess the effect of the addition of humic mineral substances from oxyhumolite to the diet of high-producing cows on the chemical composition, mineral content and rennet clotting ability of milk.

**Abstract:**

The study was conducted to determine the effect of humic mineral substances from oxyhumolite added to the diet of Holstein-Friesian cows on the coagulation properties, proximate chemical composition, and mineral profile of milk. The experiment was conducted on 64 cows divided into two groups of 32 each, control (CON) and experimental (H). The group H cows received the humic mineral substances as feed additive, containing 65% humic acids, for 60 days (100 g cow/day). Milk samples were collected twice, after 30 and 60 days. After 30 days no significant changes were observed in the chemical composition, somatic cell count (SCC), mineral content (except potassium), or curd texture parameters. However, the coagulation properties improved. The milk from group H after both 30 and 60 days coagulated significantly (15%) faster on average (*p* < 0.05), and the curd was about 36% and 28% firmer after 30 and 60 days, respectively (*p* < 0.05). After 60 days there was an increase in the content of fat (by 0.27 p.p.; *p* = 0.041), protein (by 0.14 p.p.; *p* = 0.012), and casein (by 0.12 p.p.; *p* = 0.029). SCC decreased by 20% (*p* = 0.023). The curds were significantly harder and less fracturable compared to the control. Calcium and iron content increased as well. The results indicate that humic mineral substances from oxyhumolite in the diet of cows can improve the suitability of milk for cheese production.

## 1. Introduction

Global milk production is continually growing, and a large portion of it is used for cheese production. A key element of the suitability of milk for cheese production is its coagulation properties, which are associated with factors including chemical composition (content of protein, casein, fat, and minerals) [1,2] and somatic cell count [3].

In the European Union and in many other countries around the world, commodity milk production relies mainly on Holstein-Friesian cows. Cows of this breed are distinguished by high milk yield but are often inferior to other breeds in terms of milk quality. Research indicates problems affecting the suitability of the milk of high-producing cows for cheese production, as it may have a long clotting time and produce an insufficiently firm curd [4]. Stocco et al. [1], in an assessment of the coagulation parameters of the milk of six cow breeds (Holstein-Friesian, Brown Swiss, Jersey, Simmental, Rendena, and Alpine Gray), showed that the milk of Holstein-Friesians had the slowest rennet clotting time, and the curd had the lowest firmness. As many as 23% of milk samples had a prolonged clotting time, and 8.5% of samples failed to coagulate within 30 min. Problems associated with extended clotting time reduce the value of milk for processing. To increase the quantity of milk and improve its quality, attempts are made to modify the diet of cows [5,6].

Recent years have seen growing interest in the use of humic substances in animal diets [7,8,9,10]. Humic substances (including humic and fulvic acids) are regarded as safe and natural feed additives with beneficial effects on animal welfare and on the quality of animal products. Humic substances—humic acids, fulvic acids, and humins—are natural organic substances found in soil, formed by humification of dead organic matter. A rich source of these compounds is oxyhumolite (oxidised brown coal) [11]. Humic acids are the main component of these substances. This fraction is insoluble in acidic solutions (pH < 2) but soluble in solutions with a higher pH. These acids have a high molecular weight from 5000 to 10,000 Da [12]. They have many physical, chemical, and biological properties that make them suitable for use in animal husbandry. They exhibit antioxidant and anti-inflammatory effects and support the functioning of the gastrointestinal tract of animals, accelerating their growth and at the same time improving immunity and reproduction [13]. Studies on the use of humic substances in animal diets have mainly focused on monogastric animals. The use of humic substances has been shown to positively affect the productivity and health of laying poultry, as well as the chemical composition of eggs [14]. Humic acids in the diet of piglets were shown to improve their health and weight gains [15]. Existing studies have been inconclusive when evaluating the effects of humic compounds in ruminant diet [16,17,18]. There have been few studies on the use of humic acids in the diet of dairy cows [10,19,20]. The results of the study indicate that the use of humic substances as a feed additive can provide beneficial effects on milk production traits due to their ability to modify rumen fermentation patterns, however, the mechanism of action of humic acids remains unclear. Assuming that the use of humic acids in the diet of cattle will improve utilization of nutrients from the feed and the functioning of the digestive system, including rumen metabolism, we can expect increased production of milk with a stable chemical composition. The literature lacks studies on the effect of humic substances on the suitability of milk for cheese production. The aim of the study was to assess the effect of the addition of humic mineral substances from oxyhumolite to the diet of Holstein-Friesian cows on the rennet coagulation properties, proximate chemical composition, and mineral profile of milk.

## 2. Materials and Methods

### 2.1. Animals and Diets

The study was carried out on a commercial dairy farm raising only Holstein-Friesian cows. Sixty four cows were selected from a herd of 150 heads on the basis of number of lactation (first or second) and stage of lactation (between 35th and 120th day, 75 on average). Selected cows (average weight 632 kg, milk yield 36.41 kg) were randomly allocated into two groups (*n* = 32) and fed either the control diet (CON) with no feed additive or an experimental diet (H) containing oxyhumolite additive. Two different periods of diet administration (30 and 60 days) were included. The animals were housed in a free-stall barn and fed in a partial mixed ration system. The basal control diet consisted of maize silage (63%), haylage (30%), rapeseed meal (2%), high protein concentrate (39% protein, Biofeed, 2%), soybean meal (1.6%), straw (0.9%), protect buffer (Biofeed, 0.3%), and chalk (0.2%). The HUMAC^®^ Natur AFM feed additive, containing 65% humic acids in dry matter, was added to the experimental diet in the amount of 100 g per cow per day, according to the manufacturer’s recommendations. The composition of the preparation is presented in Table 1. The basal feed was designed for milk yield of 28 kg. Per every 2.5 kg of milk above 28 kg, cows received additionally 1 kg of complete fed with 20% protein from a feed station. The feed additive was gradually added to the experimental feed for seven days: 30 g was added on the first day of the experiment, and then the amount was increased by 10 g each day until reaching the level of 100 g per cow recommended by the manufacturer. The cows were milked twice a day (at 6 a.m. and 6 p.m.) in a 20-stall herringbone milking parlour.

### 2.2. Milk Samples

Milk samples were collected twice during the experiment: once 30 days after the additive was introduced to the feed of the H group cows and again after 60 days. The research material comprised in total 128 milk samples (64 animals × 2 samples) obtained from a complete milking procedure, individually from each cow during morning milking, in the amount of 500 mL. Then the samples were transported in refrigerated conditions to the laboratory for analysis. Analysis of acidity, chemical composition and coagulation properties were performed on fresh milk, 3 h after milking was completed. The content of minerals was determined in frozen milk (−20 °C) up to 30 days after sample collection. All analyses were performed in duplicate.

### 2.3. Laboratory Analyses

#### 2.3.1. Acidity and Proximate Chemical Composition of Milk

Active acidity (pH) was determined in the milk samples using the multi-purpose pIONneer 65 Meter (Radiometer Analytical, Villeurbanne, CEDEX France), set to automatic temperature compensation and fitted with a combined electrode (E16M340) calibrated at two points with pH buffer solutions (Alfachem, Poland), i.e., pH 4.00 and pH 7.00 (±0.02 at 20 °C). Content of fat, protein, lactose, and dry matter was determined with the Infrared Milk Analyzer (Bentley Instruments, Inc., Chaska, MN, USA). Casein content was determined according to the AOAC method [21]. Somatic cell count (SCC) was determined with a Somacount 150 (Bentley Instruments, Inc., Chaska, MN, USA), and urea content with a ChemSpec analyser (Bentley Instruments, Inc., Chaska, MN, USA). Data on the cows’ daily milk yield was obtained from records (RW-2 reports) kept by the Polish Federation of Cattle Breeders and Dairy Farmers.

#### 2.3.2. Milk Coagulation Properties (MCP)

Milk coagulation properties (MCP) were analysed using the V2 Lactodynamograph (Foss Italia, Padova, Italy). A 10 mL volume of milk, heated to 35 °C, was mixed with 200 μL of rennet solution (Hansen Naturen Plus 215, Rome, Italy, diluted in distilled water to obtain a 1.2% (wt/vol) solution, with a final concentration of 0.0513 IMCU/milk mL). The properties recorded were rennet coagulation time (RCT, min), the time interval between rennet addition and gelation; curd-firming time (K_20_, min), the time between gelation and attainment of a curd firmness of 20 mm; and curd firmness at 30 min after rennet addition (A_30_, mm).

To obtain rennet curds, 50 mL of milk was placed in glass beakers and heated in a water bath to 35 °C, and then the rennet solution was added (0.0513 IMCU/milk mL). Next, the beakers were incubated for 45 min at 35 °C to obtain curd. Following incubation, the curds were cooled to ambient temperature (22 ± 1 °C) and analysed. Parameters describing curd texture (fracturability, hardness, cohesiveness, adhesiveness, springiness, gumminess, and chewiness) were determined according to Wolanciuk et al. [22] using the Zwick/Roell Proline BDO-FB0.5TS testing machine (Zwick GmbH and Co, Ulm, Germany). A cylindrical die, 45 mm in diameter and 5 mm high, penetrated the curd twice in succession to a depth of 25 mm at a speed of 1 mm/s. The compression cycles were separated by a 2 s relaxation phase. The measurements and results were recorded in TestXpert II software.

#### 2.3.3. Analysis of Mineral Content

A 5 mL volume of 65% nitric acid and 1 mL of hydrochloric acid 30% (Suprapur grade; Merck, Germany) was poured over 1.2 mL mixed samples of milk in PFA vessels (PFA). The method was verified by analysing the certified reference material ERM-BD151 skimmed milk powder (JRC, IRMM, Geel, Belgium) in exactly the same manner as the test samples. Mineralization of all solutions together with a blank sample was carried out in a MARSXpress 5 microwave digestion oven (CEM Corporation, Matthews, NC, USA) in a closed system. Mineralization consisted of three steps according to the programmed parameters of the oven:Power: 400 W/25% max power; increment: 10 min/100 °C; holding time 10 minPower: 800 W/50% max power; increment: 10 min/150 °C; holding time 5 minPower: 1600 W/100% max power; increment: 15 min/200 °C; holding time 20 min

Then the digests were transferred to volumetric flasks using ultrapure water produced in a HLP 20UV demineralizer (HYDROLAB, Poland). Schinkel buffer (enth./cont. 10 g/L CsCl + 100 g/L La; Merck, Darmstadt, Germany) was added for the analysis of Ca, Mg, K, and Na to minimize interference. Concentrations of Ca, K, Na, Mg, Zn, Fe, Mn, and Cu in the solutions were determined with the Varian AA240FS Fast Sequential Atomic Absorption Spectrometer (Varian Australia Pty Ltd., Mulgrave, Australia). The elements were atomized in the flame of a burner fed with a mixture of air (oxidizing gas, flow 13 L/min) and acetylene (combustible gas, flow 2.0 L/min). The following parameters were set for the spectrometer: instrument mode—absorbance; measurement mode—integration; calibration mode––concentration; calibration algorithm—New Rational; and replications—5 × standard/3 × sample. Analytical wavelengths (nm) were as follows: for Ca 422.7, Mg 285.2, Zn 213.9, Fe 248.3, Mn 279.5, Cu 324.7, K 766.5, and Na 589.0. Background correction (BC) was used for the determination of Mg, Zn, Fe, Mn and Cu. A standard curve was plotted for the elements using single-element standard solutions with a mass concentration of 1000 mg/L (Merck, Germany). The following detection limits (LOD) were used in the analysis: 0.01 mg/kg for Na, Zn, Mn, and Cu; 0.04 mg/kg for K, 0.09 mg/kg for Fe, 0.22 mg/kg for Ca, and 0.47 mg/kg for Mg. The content of major and minor elements in the samples was expressed in mg/L wet weight.

### 2.4. Statistical Analysis

Data were initially subjected to ANOVA using the GLM procedure using Dell Sta-tistica ver 5. 13 software [23]. Individual cow was the experimental unit, and the following model was used:y_ijk_ =μ + α_i_ + β_j_ + (αβ)_ij_ + ε_ijk_
where, μ is the overall mean; α the effect of diet (i = CON, H); β the effect of feeding period in days (j = 30, 60); αβ the interaction of diet × feeding period, and ε the error with the animal as a random factor nested in the treatment. Due to a lack of significant interaction of experimental factors two different feeding periods (30 and 60 days) were analysed separately as a main effect. The Student’s t-test was used to determine the differences between means and considered significant at *p* < 0.05. In tables the mean ± standard deviation and *p*-value are presented.

## 3. Results and Discussion

### 3.1. Milk Yield and Chemical Composition

Table 2 presents the effects of the addition of humic mineral substances to the diet of cows on milk yield and chemical composition. No changes in milk yield were observed during the experiment. After 30 days of use of the feed additive no significant changes were noted in the chemical composition or somatic cell count of the milk. After 60 days of use of the additive, a significant increase was noted in the content of dry matter components, including fat (by 0.27 p.p.; *p* = 0.041), protein (by 0.14 p.p.; *p* = 0.012), and casein (by 0.12 p.p.; *p* = 0.029).

Hassan et al. [20] added humic acids from brown clay soil to the diet of Holstein-Friesian cows in the amount of 5 and 10 g/kg diet and showed a significant increase in milk production (from 17.21 kg in the control group to 18.22 and 18.26 kg in the experimental groups receiving a diet supplemented with humic acids in the amount of 5 and 10 g/diet, respectively). No differences, however, were noted in the chemical composition of the milk. It should be highlighted that daily milk yield in present study was very high relatively (over 38 kg), indicating that evaluated cows have reached the full potential of milk production. Potůčková and Kouřimská [24] used the Humafit supplement (containing 65.34% humic acid) in the diet of dairy cows for 84 days and observed a significant (*p* < 0.05) increase in the content of crude protein and casein in the milk from day 56 of the experiment, a finding similar to the results of the present study. On day 84 of the experiment, the crude protein content of the milk of the cows in the experimental group was 0.67 p.p. higher and casein content was 0.49 p.p. higher than on day 0. No significant differences were noted in the content of dry matter, fat, or lactose in the milk. A study by Yüca and Gül [10] in which cows received a commercial preparation containing humic acids (activated leonardite in the amount of 400,000 mg/kg; the authors did not report the percentage content of humic acids) in the amount of 75 g and 150 g showed only an increase in fat content in the milk (by 0.4 p.p.), but no changes in the content of other milk components.

The most plausible explanation for increased content of protein in milk relies on the modification of the function of rumen microbiota and consequently the improvement of protein metabolism by reducing or eliminating protozoan activity. Degirmencioglu and Ozbilgin [25], who added humic acids to the diet of goats, found no changes in either the amount or the chemical composition of the milk produced. El-Zaiat et al. [26] showed that the use of humic acids in the diet of goats had a positive effect on the quantity and quality of milk produced. Animals whose diet was supplemented with humic acids produced more milk (by 18%) with higher protein content, and at the same time the concentration of protozoa in the rumen was much lower than in the control group (2.83 vs. 3.12 × 10^5^ cells/mL). The authors also observed higher fat content and a decrease in the urea level and somatic cell count in the milk of the experimental goats.

Benchaar et al. [27] and by El-Zaiat et al. [26] report that protozoa ingest and digest a large number of rumen bacteria, thereby reducing the flow of bacterial protein from the rumen to the duodenum. Due to the ability of protozoa to carry out proteolysis and deamination, a reduction in their numbers in the rumen leads to an increase in the amount of nitrogen of microbial origin reaching the duodenum. The beneficial changes in milk may thus have been due in part to better utilization of nutrients owing to the effects of humic substances. They modify the intestinal microbiome and thereby improve the utilization of nutrients from feed, which has a beneficial effect on the chemical composition of the milk produced [24]. Hassan et al. [20] showed that the addition of humic substances to the diet of cows significantly (*p* < 0.05) increased the digestibility coefficient of crude protein. Humic acids have the ability to bind nitrogen, thereby reducing its excretion. Terry et al. [18] also noted a significant increase in the digestibility of protein in the rumen of heifers as the content of humic acids in their diet increased from 0 to 300 mg per kg body weight. The level of *Firmicutes* bacteria also increased with the amount of humic substances in the diet. Protein digestibility was shown to be positively correlated with the number of these bacteria (r = 0.37, *p* < 0.05).

The urea concentration in milk can be used to assess the energy–protein balance in feed. The recommended urea level is in the range of 171–321 mg/L in the United States, 171–300 mg/L in Canada, 200–300 mg/L in France, and 180-300 mg/L in Denmark [28]. According to the Polish Federation of Cattle Breeders and Dairy Farmers (PFCBDF) [29], the optimum urea level is 180–280 mg/L, with protein content of 3.2–3.6%. In the present study, during 60-day use of the humic acid supplement in the diet of cows, the urea content in the milk was 203.54–239.68 mg/L, which is within PFCBDF recommendations and indicates a well-balanced diet.

A significant reduction in the somatic cell count (SCC) of milk was obtained in our study. After 60 days of a diet with humic acids, the SCC in the milk decreased by 20% (*p* = 0.023). Zigo et al. [19] administered the Humac Natur AFM preparation to cows for 50 days before calving and noted a significantly (*p* < 0.05) lower somatic cell count in the milk after 10 days of lactation. After 30 days of lactation, however, no differences were noted in the somatic cell count in the milk of cows from the experimental and control groups.

The somatic cell count in milk is an important parameter in the evaluation of both the health safety of milk and its suitability for processing. According to European Union regulations [30], the SSC of commodity milk for processing may not exceed 400,000/mL (geometric mean from three consecutive months). An elevated somatic cell count in milk has an adverse effect on its chemical composition (including lower content of casein and fat and higher content of whey proteins), which negatively affects its coagulation properties, thus reducing the quantity and quality of cheese produced [31]. Franceschi et al. [3] showed that the yield of Parmigiano Reggiano cheese aged for 24 months made from milk with an SSC > 400,000/mL was 9% lower than the cheese yield from milk with an SCC not exceeding 400,000/mL. A high somatic cell count in milk used for cheese production has also been linked to poorer fat retention by the curd. Therefore, a decrease in SCC in milk is beneficial both for milk producers and for dairies producing rennet cheese.

### 3.2. Milk Coagulation Properties

The coagulation properties of milk are a key parameter in the assessment of its suitability for cheese production. Milk coagulation involves a number of physicochemical changes taking place at the level of the casein micelles, which lead to the formation of curd. The course of this process is reflected in cheese production, its yield, and the quality of the final product. The main problems in cheese production are rapid coagulation of milk (in the case of milk with high acidity), late or no coagulation, poor firmness of the curd when cut, and slow syneresis [1]. The use of humic mineral substances in the diet of cows has been shown to significantly improve the rennet clotting ability of milk (Table 3). A reduction in clotting time and the time needed for the curd to attain a specified firmness, as well as an increase in firmness 30 min after application of the coagulant, was observed from day 30 of the experiment. Milk from the cows in the experimental group clotted on average 15% faster, and the curd was 36% and 28% firmer (on days 30 and 60 of the experiment, respectively), compared to the control group. We found no information in the literature on the effect of the use of humic acids in the diet of cows on parameters of the suitability of milk for cheese production. Only one paper has been published, by a Russian team of researchers, in which a shorter rennet clotting time (by 0.3–1.1 min) was observed following the use of Maks Super Gumat, containing 9.45% humic acids, in the diet of cows [32]. However, the authors did not provide details such as the number of cows, their diet, the amount of the additive used, or the method of determining clotting time. In the authors opinion, the improvement of milk coagulating properties resulted from both an increase in the protein content of milk (including casein) and fat, as well the mineral content (mainly calcium). In a previous study, Teter et al. [33] showed that the content of protein, casein, and fat was positively correlated with the firmness of the curd after 30 min (r = 0.273, *p* ≤ 0.001, r = 0.287, *p* ≤ 0.001, and r = 0.136, *p* ≤ 0.01, respectively), but negatively correlated with the curd-firming time (r = −0.339, *p* ≤ 0.001, r = −0.323, *p* ≤ 0.001, and r = −0.236, *p* ≤ 0.001, respectively). In turn, the calcium content was correlated negatively with rennet clotting time and curd firming time (r = −0.284, r = −0.275, *p* ≤ 0.01, respectively), but positively with curd firmness (r = 0.301, *p* ≤ 0.001) [2]. Therefore, it can be concluded that the improvement in the coagulating properties of milk is the result of the multiple effects of different humic substances on the cow’s organism and milk secretion.

Table 4 presents the texture parameters of the rennet curds. In the first period of the experiment (after 30 days of use of the feed additive) no significant differences were noted in the parameters, although an upward trend in the hardness of curds and downward trends in fracturability and springiness were observed. After 60 days of the experimental diet, the rennet curds obtained from the milk of cows receiving humic substances were shown to be significantly harder (*p* = 0.037) than the curds from the control group (4.36 and 3.56 N) and less fracturable (*p* = 0.042) – 3.60 and 4.18 N. The trend of decreasing springiness persisted. The texture parameters of curds are very important for mechanical processing, and ultimately for the quantity and quality of cheese. Cutting of curd with appropriate hardness and low fracturability can limit losses of curd fines and nutrients in the whey. If the curd is excessively springy when cut (cut too late), prolonged syneresis results in higher water content in the cheese and undesirable texture parameters [34]. We found no information in the literature about the effects of the use of humic acids in the diet of cows on the texture parameters of rennet curds. Previous research by Teter et al. [33] showed highly significant relationships between milk coagulation parameters and the texture of the rennet curds obtained. Curd fracturability was shown to be correlated negatively with rennet coagulation time (RCT) (r = −0.457; *p* < 0.001) and curd firming time (K_20_) (r = −0.472; *p* < 0.001) and positively with firmness at 30 min after addition of the enzyme (A_30_) (r = 0.684; *p* < 0.001). Therefore, the shorter coagulation time and shorter time needed for the curd to attain a specified firmness had a positive effect by reducing its fracturability. 

The changes in curd texture parameters shown in the present study may be related to conditions of the rennet coagulation process, which in turn was modified by differences in the concentrations of crude protein, casein, and fat.

### 3.3. Mineral Profile of Milk

Table 5 presents the effects of the use of the humic mineral additive in the diet of cows on the content of selected minerals in the milk. After 30 days of use of the additive, a significant (*p* = 0.001) decrease (7%) in potassium was noted in the milk (by 117.7 mg/kg). The concentration of calcium was slightly (not statistically) increased (by 5%), whereas contents of sodium, magnesium, iron, manganese, and copper were similar. After 60 days of the experiment, no differences were noted in the potassium content in the milk, but there was a significant increase in the content of calcium (by 13%; *p* < 0.001) and iron (by 34%; *p* = 0.004) in the milk of cows in the experimental group. A decrease in potassium content should not be considered negatively in terms of the nutritional value of the milk or the mineral status of the cows. It is commonly assumed that NPK fertilisation of crops results in sufficient, and sometimes even excessive, potassium in the ration for cows and therefore there is no need to supplement it. Particularly as potassium has a high bioavailability, which contributes to its higher availability in the organism. Fadlalla et al. [35] indicates a negative correlation coefficient for potassium and calcium contents (r = − 0.39; *p <* 0.05), which may partly explain the results obtained.

Calcium is one of the most valuable minerals in milk in terms of both nutritional value and suitability for processing. It is much more easily absorbed than calcium from plant products. Adequate calcium intake in the diet decreases the risk of osteoporosis and colorectal cancer. Calcium, magnesium, and potassium are responsible for regulation of blood pressure. About 65% of calcium is bound to casein micelles, which play a key role in curd formation [36].

The literature lacks information on the effect of humic acids on the content of minerals in milk. Skalická et al. [37] added the Humac Natur preparation to the diet of chickens and showed a significant increase in calcium, iron, and aluminium in their serum, liver, and muscles. Yüca and Gül [10] used a humic acid additive in the diet of cows before and after calving and observed a significant increase in the level of calcium in the blood after calving. No differences were observed in magnesium and phosphorus levels. The results indicate that humic substances, as the carriers of minerals in chelate form, in the diet of animals can affect mineral metabolism, but further research is needed to explain this mechanism

## 4. Conclusions

Organic-mineral substances containing humic acids are becoming an increasingly common natural additive in animal diets due to properties that promote health, support digestion, and improve production efficiency. The effects of their use in the diet of dairy cattle on the quantity and quality of milk are unclear. Previous literature results are difficult to compare due to differences in the experiments conducted, i.e., in the sources of humic acids, the doses applied, and the time of their application.

Our results indicate that the use of an oxyhumolite derived humic substance containing 65% humic acids and minerals in an easily digestible form in the amount of 100 g per cow per day did not affect the cows’ productivity. However, it had a beneficial effect by increasing the content of protein, casein, and fat in the milk and reducing the SCC, but only after 60 days of use. After this period, higher contents of calcium and iron were observed in the milk as well. At this time there was also an improvement in texture parameters (higher hardness and lower fracturability values) of the rennet curds obtained from the milk. Beneficial changes in milk coagulation parameters (shorter coagulation time, shorter time for the curd to attain a specified firmness, and increased curd firmness), were observed after 30 days of use of the additive and persisted to the end of the experiment.

The results of the study indicate that the suitability of milk for cheese production can be improved by introducing a humic mineral additive to the diet of cows. Further studies should verify whether modification of milk coagulation properties influences cheese production processes and the quantity and quality of the final product.

## Figures and Tables

**Table 1 animals-11-01970-t001:** Chemical composition of HUMAC^®^ Natur AFM feed additive (content in dry matter).

Component	Content
Humic acids	65%
Fulvic acids	5%
Calcium	42,278 mg/kg
Magnesium	5111 mg/kg
Iron	19,046 mg/kg
Copper	15 mg/kg
Zinc	37 mg/kg
Manganese	142 mg/kg
Cobalt	1.24 mg/kg
Selenium	1.67 mg/kg
Vanadium	42.1 mg/kg
Molybdenum	2.7 mg/kg

**Table 2 animals-11-01970-t002:** Effect of the addition of oxyhumolite to the diet on yield and milk parameters (Mean ± SD).

Parameter	After 30 Days			After 60 Days		
	CON	H	*p*	CON	H	*p*
Milk yield (kg)	38.51 ± 6.79	38.47 ± 6.40	0.978	38.65 ± 6.20	39.46 ± 6.70	0.647
pH	6.78 ± 0.05	6.79 ± 0.06	0.281	6.76 ± 0.06	6.79 ± 0.07	0.256
Fat (%)	3.67 ± 0.50	3.84 ± 0.51	0.173	3.66 ± 0.48	3.93 ± 0.46	0.041
Protein (%)	3.27 ± 0.23	3.35 ± 0.20	0.173	3.30 ± 0.18	3.44 ± 0.21	0.012
Casein (%)	2.60 ± 0.20	2.67 ± 0.19	0.204	2.61 ± 0.17	2.73 ± 0.18	0.029
Lactose (%)	4.87 ± 0.16	4.83 ± 0.15	0.332	4.82 ± 0.14	4.76 ± 0.16	0.066
Dry matter (%)	12.51 ± 0.96	12.76 ± 0.59	0.058	12.47 ± 0.63	12.85 ± 1.04	0.031
Urea (mg/kg)	203.54 ± 52.60	225.14 ± 32.87	0.070	210.88 ± 42.53	239.68 ± 51.41	0.097
SCC (thous./mL)	258.88 ± 50.35	225.83 ± 75.62	0.271	261.08 ± 49.78	208.45 ± 40.17	0.023

CON—control group; H—experimental group; SCC—somatic cell count; *p*—value.

**Table 3 animals-11-01970-t003:** Effect of the addition of oxyhumolite to the diet on milk coagulation parameters (Mean ± SD).

Parameter	After 30 Days			After 60 Days		
	CON	H	*p*	CON	H	*p*
RCT (min)	20.86 ± 4.84	17.78 ± 4.81	0.025	20.99 ± 4.58	18.01 ± 3.34	0.049
A_30_ (mm)	19.01 ± 5.33	25.86 ± 6.48	0.013	18.70 ± 8.04	23.94 ± 6.79	0.039
K_20_ (min)	7.76 ± 1.98	5.49 ± 2.55	0.007	7.71 ± 1.86	5.84 ± 1.35	0.009

CON—control group; H—experimental group; RCT—rennet coagulation time; A_30_—curd firmness 30 min after rennet addition; K_20_—curd-firming time; *p*—value.

**Table 4 animals-11-01970-t004:** Effect of the addition of oxyhumolite to the diet on rennet curd texture parameters (Mean ± SD).

Parameter	After 30 Days			After 60 Days		
	CON	H	*p*	CON	H	*p*
Fracturability (N)	3.60 ± 1.15	3.75 ± 1.18	0.649	3.60 ± 0.81	4.18 ± 0.88	0.042
Hardness (N)	3.62 ± 1.12	3.84 ± 1.22	0.684	3.56 ± 1.08	4.36 ± 0.99	0.037
Adhesiveness (N)	1.69 ± 0.65	1.91 ± 0.94	0.400	1.62 ± 0.51	2.03 ± 0.57	0.144
Springiness	2.06 ± 0.93	1.59 ± 0.69	0.067	2.02 ± 0.84	1.59 ± 0.59	0.163
Guminess (N)	0.42 ± 0.09	0.44 ± 0.11	0.412	0.43 ± 0.08	0.46 ± 0.09	0.384
Chewiness (N)	0.79 ± 0.16	0.75 ± 0.14	0.359	0.89 ± 0.61	0.74 ± 0.31	0.438
Cohesiveness (mJ)	0.12 ± 0.04	0.12 ± 0.03	0.979	0.13 ± 0.05	0.10 ± 0.02	0.061

CON—control group, H—experimental group; *p*—value.

**Table 5 animals-11-01970-t005:** Effect of the addition of oxyhumolite to the diet on the mineral profile of milk (Mean ± SD).

Parameter	After 30 Days			After 60 Days		
	CON	H1	*p*	CON	H2	*p*
Ca (mg/kg)	1113.13 ± 82.39	1174.40 ± 70.33	0.867	1104.84 ± 61.69	1253.23 ± 99.81	0.000
K (mg/kg)	1589.48 ± 105.08	1477.78 ± 118.23	0.001	1539.98 ± 114.62	1524.24 ± 80.34	0.700
Na (mg/kg)	356.29 ± 27.37	353.62 ± 86.16	0.629	348.69 ± 44.84	387.91 ± 130.29	0.335
Mg (mg/kg)	108.21 ± 10.23	105.18 ± 8.07	0.459	107.06 ± 8.44	109.79 ± 8.19	0.452
Zn (mg/kg)	4.48 ± 0.98	4.22 ± 0.76	0.476	4.56 ± 0.98	4.33 ± 0.58	0.489
Fe (mg/kg)	0.33 ± 0.12	0.41 ± 0.08	0.106	0.32 ± 0.09	0.43 ± 0.08	0.004
Mn (mg/kg)	0.05 ± 0.002	0.06 ± 0.003	0.118	0.05 ± 0.01	0.07 ± 0.01	0.065
Cu (mg/kg)	0.05 ± 0.007	0.03 ± 0.017	0.341	0.06 ± 0.004	0.05 ± 0.003	0.504

CON—control group; H—experimental group; *p*—value.

## Data Availability

The data presented in this study are available on request from the corresponding author.

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
