# Peer review of "The Effect of Humic Mineral Substances from Oxyhumolite on the Coagulation Properties and Mineral Content of the Milk of Holstein-Friesian Cows"

_animals, 2021, doi:10.3390/ani11071970_

Round 1

Reviewer 1 Report

Teter et al. have fed a commercial feed additive (HUMAC® Natur AFM) to dairy cows, that contain humic substances and many other minerals. Henceforth, authors have collected milk samples after 30 or 60 days and composition and texture properties were subsequently studied. Authors have observed the group provided with feed additive coagulate faster and form a firmer gel, among some notable milk compositional changes. Authors claim this to be a result of humic acids, which is accounted for 65% of humic acid (Table 1). Hence, my opinion is, it's not fair to formulate the title as  “The effect of humic acids on…”, when the humic acid is just a component of a commercially formulated feed additive. However, the author's discussion is not supportive of this observation and I have the suspicions, these observed milk coagulation differences are mainly associated with the minerals, especially the high amount of intake of calcium with the H group compared to the control group. The authors have not taken this into account when discussing it. I would like to see this, and the authors justifications on how differences in minerals affect these observed variations as well as the exact effect of humic acid (e.g. how it is affected? In which path? Any hypothesis or facts based on literature). I wish to request authors to reformulate their discussion to provide a balanced overview, identify its limitation, and associated benefits. The bottom line is, simply authors can not claim all the observed differences are due to humic acid in feed additive. If so, a strong foundation and explanations are needed.

In addition to this, much attention must be drawn to correct English syntax and grammar. Paper contain only tables, which might be a good idea to include a figure/graph to avoid the monotonous nature. Perhaps, this article is at best, suitable for a technical note, after correcting the points highlighted above.

Author Response

Dear Reviewer,

Thank you for giving us the opportunity to submit a revised draft of our manuscript to Animals.

We appreciate the time and effort that you have dedicated to providing your valuable feedback on our manuscript. We deeply appreciate any suggestions or notice of weaknesses to improve the quality of our article. The manuscript was proofread by Ms. Sara Wild (native speaker). Please find below the answers to your comments.  

In recent years there has been a growing interest in products that are sources of humic substances.

Humic acids are their main constituent, but in addition there are fulvic acids and mineral compounds in chelated form, among others. I fully agree that the potential effects of humic substances are not due solely to the properties of humic acids but to the complex action of many compounds, including minerals. Thank you for highlighting such an important aspect that was not sufficiently discussed in the paper submitted for evaluation. The title of the manuscript has been corrected. The humic compounds used in the experiment were derived from oxyhumolite, which is valued for its high humic acid content. The results of studies conducted on different animal species indicate that the use of humic compounds in their nutrition can positively affect production effects. However, the effects of their application are not always clear, and the mechanism of humic acids' action is not fully explained. On the basis of literature data, it can be concluded that production effects depend largely on the applied source of humic compounds (different chemical composition), dose, and time of application. In the case of dairy cows, the reason for undertaking the research was the information on the possibility of modifying rumen metabolism, composition of rumen microflora, which in turn translates into feed utilisation, increased protein digestibility and more favourable chemical composition of milk. In assessing the suitability of milk for cheese production, the content of milk protein is important, particularly casein. The ability of milk to coagulate and the quality of the formed casein curd largely depend on these components. When analysing the enzymatic coagulation process, the importance of minerals, particularly calcium, cannot be ignored. It is well known that the calcium content in milk is correlated with a shorter milk coagulation time and a higher curd firmness. Analysing the obtained results of the study it can be observed that already after 30 days of the experiment the coagulating properties of milk improved significantly, while the calcium content did not change significantly. At this stage of the study it is not possible to state unequivocally which element of such a complex as milk mainly improved the coagulating properties of the raw material. In authors opinion, it should be assumed that the demonstrated improvement of milk quality is a result of changes in the basic chemical composition, mineral profile of milk, but also attention should be paid to the decrease in the level of somatic cells. In turn, the observed changes are a result of complex action of humic substances being a carrier of mineral compounds in an easily assimilable form. The research undertaken was preliminary in nature. Their aim was to verify the hypothesis on the possibility of improving the technological suitability of milk by applying natural humic substances. After obtaining positive results, further research should be conducted in order to understand the mechanisms of action of these substances in the animal organism, to obtain clear answers as to the role of individual components. We believe that you will accept our explanation and find this revision satisfactory.

Reviewer 2 Report

This paper researched the effect of humic acids on the milk yields and  rennet clotting of milk. The research is interesting and usefule, however, paper still need revised before it could be cosider for accepted. 

  1. The title of the manuscript was not clearly presented how to supplement the humic acids and needs to be revised. And I think it is hard to say the effects of humic acids because its only 65% in the additive?
  2. Line13-14: .......... improve the health and milk quality of cows.
  3. Line21: effects
  4. Line 58-59: added the references
  5. Line 77-82: so confused, should be more logically. In addition, this paragraph should first introduce the properties of humic substances and its researches as an animal additive (including monogastric and ruminant as you stated) . Then point out the limitations on the dairy cows or milk quality, and finally write the aim of this study. Meanwhile, this paragraph needs to supplement more references
  6. Line91-92: supplemented the detailed information of cows in each group (parity, milk yield.....). Did there any significant differences between two groups? And what the experiment design Randomized block design?
  7. Line 182-184: too simple. And do you consider the effects of time, cow factors in this analysis model? if you uesd the Randomized block design in the current study , it should be applied a MIXED model which consider these factors to analyzed the results.
  8. Line188: ......change.....was. Check through out the manuscript
  9. Line 190-193: too confused, should be more logically and concise
  10. Line 196-262: Many literatures are cited here, but the comparison and discussion of the differences among these studies and the current study are lacking . Therefore, author should be discussed deeper about the results and the possible cause . The same problem was also existed in the following part. That should be revised.
  11. For the K, significant difference was presented after 30 days but no significant differences were observed after 60 days. Why? That should be discussed in the manuscript.
  12. For the conclusion part, Many of the descriptions are similar to the INTRODUCTION and RESULTS part, that need to addressed and should more concise.

Author Response

Dear Reviewer,

Thank you for giving us the opportunity to submit a revised draft of our manuscript to Animals.

We appreciate the time and effort that you have dedicated to providing your valuable feedback on our manuscript. We deeply appreciate any suggestions or notice of weaknesses to improve the quality of our article. Please find below the line-by-line answers to your comments and concerns. We believe that you will accept our explanation and find this revision fully satisfactory.

This paper researched the effect of humic acids on the milk yields and  rennet clotting of milk. The research is interesting and usefule, however, paper still need revised before it could be consider for accepted. 

Ans. Thank you kindly for your comment.

  1. The title of the manuscript was not clearly presented how to supplement the humic acids and needs to be revised. And I think it is hard to say the effects of humic acids because its only 65% in the additive?

Ans. Thank you for pointing out this important issue. The title has been corrected. In the authors' opinion it is more appropriate to use the term humic - mineral substances from oxyhumolite instead of humic acids. Humic acids are the main constituent of oxyhumolite, but fulvic acids and minerals are also present.

  1. Line13-14: .......... improve the health and milkquality of cows.

Ans. Corrected.

  1. Line21: effects

Ans. The term "effects" has been replaced by "results".

  1. Line 58-59: added the references

Ans. References were added.

  1. Line 77-82: so confused, should be more logically. In addition, this paragraph should first introduce the properties of humic substances and its researches as an animal additive (including monogastric and ruminant as you stated). Then point out the limitations on the dairy cows or milk quality, and finally write the aim of this study. Meanwhile, this paragraph needs to supplement more references

Ans. The paragraph has been re-written as suggested; references have been added.

  1. Line91-92: supplemented the detailed information of cows in each group (parity, milk yield.....). Did there any significant differences between two groups? And what the experiment design Randomized block design?

Ans. The study was randomised. Cows were unified in terms of parity, lactation stage, body weight and milk yield, and then randomly allocated to 2 equal groups. The materials and methods section has been supplemented with this information.

  1. Line 182-184: too simple. And do you consider the effects of time, cow factors in this analysis model? if you uesd the Randomized block design in the current study , it should be applied a MIXED model which consider these factors to analyzed the results.

Ans. More details regarding the statistical analysis of the obtained results have been added. All indicated factors were considered in the statistical model.

  1. Line188: ......change.....was. Check throughout the manuscript

Ans. Checked.

  1. Line 190-193: too confused, should be more logically and concise

Ans. Corrected.

  1. Line 196-262: Many literatures are cited here, but the comparison and discussion of the differences among these studies and the current study are lacking. Therefore, author should be discussed deeper about the results and the possible cause. The same problem was also existed in the following part. That should be revised.

Ans. Corrected as suggested.

  1. For the K, significant difference was presented after 30 days but no significant differences were observed after 60 days. Why? That should be discussed in the manuscript.

Ans. Information on this topic has been added in the text.

  1. For the conclusion part, Many of the descriptions are similar to the INTRODUCTION and RESULTS part, that need to addressed and should more concise.

Ans. Corrected as suggested.

Reviewer 3 Report

This manuscript by Teter and colleagues have determined the effect of humic acids on the coagulation properties and mineral content of the milk of Holstein-Friesian cows. This is an interesting topic. The experimental designed well and enough samples size used in the current study. The writing is also fine. The manuscript could be published after the following revision.

  1. Abstract: “HUMAC Natur AFM feed additive” is a commercial name. Please avoid to use this in the paper, except the material section.
  2. Abstract: please add the changes/degree (percent of the changes) by the treatment with the P values.
  3. Line 107, Table 1, tables should be the form of three line. Please check all the tables and make them uniformed.
  4. Table 2, please add the detailed information about the P value for the significant (P < 0.05?). Also, please specified that the values are expressed as mean ± SD or SE? Please check these information for all the tables.
  5. Please check all the abbreviations and make sure the full name should be written when it was first appeared in the text.
  6. Line 184, the “p < 0.05” should be written as “P < 0.05”, the P should be written as capital and italic. The space need to be added before and after the “<”.
  7. Line 309, r=0.273, space needed before and after “=”. Please check throughout the paper.
  8. Line 327, please use P< 0.001 to substitute “p=0.000”.

Author Response

Dear Reviewer,

We would like to thank you for the effort that you put in reviewing our work, for its favorable opinion. We deeply appreciate any suggestions or notice of weaknesses to improve the quality of our article. All comments have been taken into account in the revised manuscript. We believe that you will find this revision fully satisfactory.

This manuscript by Teter and colleagues have determined the effect of humic acids on the coagulation properties and mineral content of the milk of Holstein-Friesian cows. This is an interesting topic. The experimental designed well and enough samples size used in the current study. The writing is also fine. The manuscript could be published after the following revision

Ans. Thank you kindly for your positive comment.

  1. Abstract: “HUMAC Natur AFM feed additive” is a commercial name. Please avoid to use this in the paper, except the material section.

Ans. The commercial name of the product has been changed to humic-mineral substances. The commercial name has only been given in the material and methods section, as suggested.

  1. Abstract: please add the changes/degree (percent of the changes) by the treatment with the P values.

Ans. In the Authors' opinion, presenting the degree of changes in milk components as a percentage difference may be confusing for readers. For example, the fat content of milk after 60 days of the experiment increased by 0.27p.p., so the difference was 7%. The Authors would have preferred to present the results in their current form. P-values were added, as suggested.

  1. Line 107, Table 1, tables should be the form of three line. Please check all the tables and make them uniformed.

Ans. Corrected.

  1. Table 2, please add the detailed information about the P value for the significant (P < 0.05?). Also, please specified that the values are expressed as mean ± SD or SE? Please check these information for all the tables.

Ans. P < 0.05 value was considered statistically significant for differences. The results in the tables are presented as mean values and standard deviation (Mean ± SD). This information is provided next to the title of each table.

  1. Please check all the abbreviations and make sure the full name should be written when it was first appeared in the text.

Ans. Corrected.

  1. Line 184, the “p < 0.05” should be written as “P < 0.05”, the P should be written as capital and italic. The space need to be added before and after the “<”.

Ans. Corrected.

  1. Line 309, r=0.273, space needed before and after “=”. Please check throughout the paper.

Ans. Corrected.

  1. Line 327, please use P< 0.001 to substitute “p=0.000”.

Ans. Corrected.

Round 2

Reviewer 1 Report

Thank you for revising the manuscript as requested and I congratulate the authors for this nice paper! Well done!

Reviewer 2 Report

All the problem and question is revised, this paper is could be consider to be accepted.